# Model-Based Reinforcement Learning via Imagination with Derived Memory

**Yao Mu**
The University of Hong Kong
muyao@connect.hku.hk

**Yuzheng Zhuang** *
Huawei Noah's Ark Lab
zhuangyuzheng@huawei.com

**Bin Wang**
Huawei Noah's Ark Lab
wangbin158@huawei.com

**Guangxiang Zhu**
Tsinghua University
guangxiangzhu@outlook.com

**Wulong Liu**
Huawei Noah's Ark Lab
liuwulong@huawei.com

**Jianyu Chen**
Tsinghua University
jianyuchen@tsinghua.edu.cn

**Ping Luo**
The University of Hong Kong
pluo@cs.hku.hk

**Shengbo Eben Li**
Tsinghua University
lishbo@tsinghua.edu.cn

**Chongjie Zhang**
Tsinghua University
chongjie@tsinghua.edu.cn

**Jianye Hao**
Huawei Noah's Ark Lab
haojianye@huawei.com

## Abstract

Model-based reinforcement learning aims to improve the sample efficiency of policy learning by modelling the dynamics of the environment. Recently, the latent dynamics model has been further developed to enable fast planning in a compact space. It summarizes the high-dimensional experiences of an agent, which mimics the memory function of humans. Learning policies via imagination with the latent model shows great potential for solving complex tasks. However, only considering memories from the true experiences in the process of imagination could limit its advantages. Inspired by the memory prosthesis proposed by neuroscientists, we present a novel model-based reinforcement learning framework called Imagining with Derived Memory (IDM). It enables the agent to learn policy from enriched diverse imagination with prediction-reliability weight, thus improving sample efficiency and policy robustness. Experiments on various high-dimensional visual control tasks in the DMControl benchmark demonstrate that IDM outperforms previous state-of-the-art methods in terms of policy robustness and further improves the sample efficiency of the model-based method.

## 1 Introduction

Model-based Reinforcement Learning (MBRL) approaches benefit from the knowledge of a model, allowing them to summarize the agent's past experiences and foresight the future outcomes in a compact latent space [1, 2, 3, 4]. Previous approaches can be generally categorized into four branches, including 1) Dyna-style algorithms, 2) sampling-based methods, 3) value expansion, and 4) backpropagation through paths. First, the Dyna-style algorithms [5] generate virtual interaction

---

*Yuzheng Zhuang is the corresponding author. Yao Mu conducted this work during the internship in Huawei Noah's Ark Lab.

data (*e.g.* ME-TRPO [6], MBPO [7]) to improve policy optimization. Second, the sampling-based methods, such as [8, 9, 10], choose action by planning or shooting (*e.g.* PlaNet [8], PETS[9]). Third, the value expansion approaches, such as [11, 12], use model-augmented rollouts to improve the estimation accuracy of cumulative returns. Finally, the methods of backpropagation through paths [13, 14] employ the analytic gradient of state value through the dynamic transition to learn the policy.

For the tasks with high-dimensional sensory inputs, backpropagation through paths is more computationally efficient compared with conventional planning algorithms, which generated numerous rollouts for the best action sequence selection procedure. Dreamer [4], as a landmark of such methods, achieves state-of-the-art (SOTA) performance on visual control tasks. It summarizes the agent's high-dimensional experiences into a compact latent space by the dynamic model, which is analogized to the way human memories are stored, and benefits behaviours learning by latent imaginations. However, recent breakthroughs based on such a method focus on optimizing the policy on imaginary trajectories with only original memory on the agent's real experiences and leave the diversity of imagination largely unstudied, resulting in noise-sensitive policy and inefficient learning. Recently, neuroscientists demonstrated that "memory prosthesis", which transforms the brain's activity patterns to the electrode signals to stimulate the human's hippocampus [15, 16], could facilitate humans memory augmentation via an external implant. Inspired by such insight, our idea is to improve behaviour learning by constructing a "memory prosthesis" with externally derived memory [17] that each signal corresponds to a state transition sequence, which can enrich the diversity of the imagination without any interaction with the real environments. We propose a novel model-based reinforcement learning algorithm, called **I**magining from **D**erived **M**emory (IDM), which aims to improve policy robustness and sample efficiency by the enriched and reliable imagination. In order to generate diverse derived memory, transformations are performed on states of the latent trajectories, which are sampled from the experiences. The agent learns to obtain future rewards by the model-based imagination from both the derived and the original memories. The optimal policy is optimized by a novel actor-critic algorithm, where the state values optimize Bellman consistency for imagined rewards, and the policy maximizes the state values by propagating their analytic gradients back through the dynamics.

In this paper, we aim to improve the diversity of imagination for model-based policy optimization with the derived memory. Our main contributions are listed below.

- We design an effective memory generator that simultaneously achieves the imagination diversity as well as the relevancy between the derived memory and the real physical states, which avoids the distortion after decoding.

- We propose a model-based policy optimization framework, unifying the analytical gradients of the imagination from both derived and original memories with learnable confidence. We also provide the upper bound of the value estimation error in the frame of IDM. This sheds light on further improvement of model-based reinforcement learning in the future.

- Experiments on DMControl [18] tasks demonstrate that IDM outperforms existing SOTA approaches in terms of robustness to uncertainty and further improves the sample efficiency of the model-based method. Ablation experiment results verify the superiority of IDM.

## 2 Preliminaries

### 2.1 Model-based Reinforcement Learning

Model-based reinforcement learning aims at optimizing a policy to maximize the cumulative rewards, by accessing to a (known or learned) model of the environment. We denote a time step as $t$, a state at $t$ as $s_t \in S$, action at $t$ as $a_t \in A$, reward function $r(s_t, a_t)$, policy $\pi_\varphi(s_t)$ and a world model $p_\theta$ to characterize the process of interacting with the environment. The goal is to find a policy parameter $\varphi^*$, which maximizes the long-horizon summed discounted rewards represented by a parameterized value function, $v_\psi(s_t) \doteq \mathbb{E}\left(\sum_{i=t}^{\infty} \gamma^{i-t} r_i\right)$ with parameter $\psi$ and the discounted factor $\gamma$.

### 2.2 World Model

The world model characterizes the process of interaction with the environment. The recurrent stochastic state model (RSSM), which is used in PlaNet and Dreamer, is the SOTA solution for building latent dynamics [8]. RSSM can facilitate long-term predictions and enable the agent to

imagine thousands of trajectories in parallel. It can represent both the historical memory and the dynamic uncertainty by using a recurrent neural network.

Let us denote the sequence of observations as $\tilde{o} = \{o_0, o_1, \ldots, o_T\}$, action sequence as $\tilde{a} = \{a_0, a_1, \ldots, a_T\}$, rewards as $\tilde{r} = \{r_0, r_1, \ldots, r_T\}$, and the corresponding latent states $\tilde{z} = \{z_0, z_1, \ldots, z_T\}$. RSSM assumes that the latent state $z_t$ consists of $z_t = (s_t, h_t)$, where $s_t, h_t$ are the probabilistic and deterministic variables respectively. RSSM consists of four key components, including (1) the representation model $p(z_t|z_{t-1}, a_{t-1}, o_t)$, which encodes observations and actions to create continuous vector-valued latent states with Markovian transitions, (2) the transition model $q(z_t|z_{t-1}, a_{t-1})$, which is built by a recurrent neural network, and is to predict the future latent states with historical memory and actions, (3) the observation model $q(o_t|z_t)$, which is used to provide a reconstruction learning signal, and (4) The reward model $q(r_t|z_t)$ predicts the rewards given to the latent states. We use $p$ to denote distributions that generate samples in the real environment and $q$ to denote their approximations that enable latent imagination. Specifically, the transition model allows us to predict ahead in the compact latent space without any interaction in the real environment.

# 3 Robust World Model for Derived Memory Generation

Based on RSSM, diverse memory vectors could be generated intuitively by transforming the original memories while putting forward higher requirements on the world model's robustness to the latent space noise. Because if the derived memory can not correspond to the real physical state, i.e., the reconstructed image is distorted, it could not effectively enrich the imagination but bring disturbance for policy learning. As shown in the second row in Figure 1a and Figure 1b, if we add an random noise to the latent state $z_t = (h_t, s_t)$, the derived latent state $z'_t = (h'_t, s'_t)$ in RSSM results in low-fidelity reconstructed images, which could not be mapped to the real physical states.

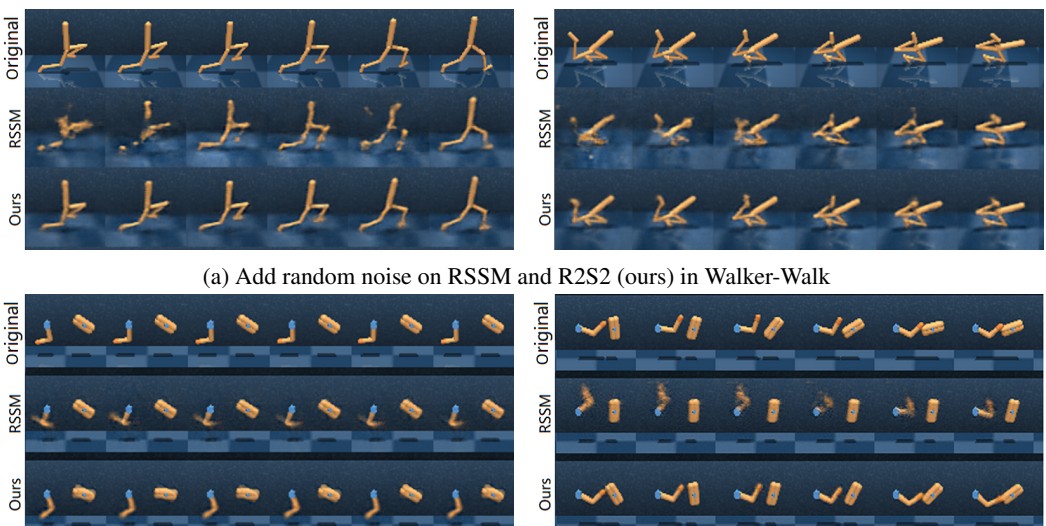

(a) Add random noise on RSSM and R2S2 (ours) in Walker-Walk

(b) Add random noise on RSSM and R2S2 (ours) model in Finger-Spin

Figure 1: Comparison of reconstruction results of the derived latent states with different world models. The first row is the original image inputs, the second row is the images decoded from the derived latent states by RSSM, the third row is the images decoded from the derived latent states by R2S2 model (ours). (Note that the shown results are randomly selected from test episodes, and the derived latent states are generated by adding Gaussian noise obeys $\mathcal{N}(0, 0.5)$ to each dimension of $(h_t, s_t)$).

To alleviate this issue, we propose a Robust Recurrent Stochastic State (R2S2) model to improve the latent model's robustness. We add a decoding constraint to the RSSM, which improves relevancy between the derived latent states and the real physical states, thus avoiding the distortion after decoding. Denote $\xi_h$ is a noise that obeys $N(0, \sigma_h)$, $h'_t$ is the derived hidden state which adds the noise $\xi_h$ to the hidden state $h_t$, $\xi_s$ is a noise that obeys $N(0, \sigma_{s'})$, $s'_t$ is the derive stochastic

state which adds the noise $\xi_s$ to $s_t$, $o_t$ is the reconstructed image from $z_t = (h_t, s_t)$, and $o'_t$ is the reconstructed image from $z'_t = (h'_t, s'_t)$.

The components of the R2S2 are optimized jointly to increase the variational lower bound [19] with a decoding constraint. The variational lower bound $J_M$ includes reconstruction terms for observations $J_O^t$ and rewards $J_R^t$ and a KL regularizer $J_D^t$ [4]. The expectation is taken under the dataset and representation model,

$$\min \mathcal{J}_M = \mathbb{E}_p \left( \sum_t \left( J_O^t + J_R^t + J_D^t \right) \right) \quad \text{s.t.} D_{\text{KL}} \left\{ p(o'_t|z'_t), p(o_t|z_t) \right\} \leq \delta \tag{1}$$

$$J_O^t = \log q(o_t|z_t) \quad J_R^t = \log q(r_t|z_t) \quad J_D^t = -\beta D_{\text{KL}} \left\{ p\left(z_t|z_{t-1}, a_{t-1}, o_t\right), q\left(z_t|z_{t-1}, a_{t-1}\right) \right\}.$$

As demonstrated in Figure 1, our proposed R2S2 model generates high-fidelity reconstructions, which suggests the enriched dynamic representation is capable for facilitate behaviour learning. The decoding constraint leaves a margin $\delta$ between the derived branch's output and the original branch's output. The constrained problem is solved by the fixed penalty method [20] where the penalty coefficient is given in the appendix A.4. As shown in the third row of Figure 1a and Figure 1b, after adding noise on the original latent state $z_t = (h_t, s_t)$ in R2S2 model, the reconstructed images with derived latent state $z'_t = (h'_t, s'_t)$ are sharper than RSSM, and are demonstrated different states of motion compared to the the reconstructed images with original latent state $z_t = (h_t, s_t)$. In summary, the decoding constraint used in the R2S2 model aims to guide the derived memory in latent space to meet a basic condition that the neighbouring latent states from real data should also follow the real physical states.

## 4 Policy Optimization with Derived Memory

By leveraging the R2S2 model, we propose an **I**magining with **D**erived **M**emory (IDM) algorithm that aims to enable the agent to learn the policy with diverse imagination. As shown in Figure 2, the derived memory is generated by the transformation from the original memory. The agent imagines the future states and rewards with both derived and original memories. The policy is improved by the enriched imagination with prediction reliability weight under the actor-critic framework.

### 4.1 Imagining with Derived Memory

We add several zero-mean noises $\xi_h \sim \mathcal{N}(0, \sigma_h^2)$ on the hidden state $h_t$ and $\xi_s \sim \mathcal{N}(0, \sigma_{s'}^2)$ on the stochastic state $s_t \sim \mathcal{N}(\mu_s, \sigma_s^2)$ in the original memory to extend the agent's memory which includes diverse embedded historical information. Thus the agent could learn behaviours with the trajectories in an enlarged latent space, rather than just the agent's real experiences. The Gaussian noise in latent space could be theoretically mapped to any type of uncertainty in image level, and this principle is extensively used in VAE and GAN. Adding Gaussian perturbations should be an intuitive way to simulate the influences in latent space caused by the uncertainty of input image, since the stochastic part of latent state $s_t$ in the R2S2 model obeys Gaussian distribution, any image will be encoded to a Gaussian variable in latent space with R2S2 model. The latent trajectories are imagined from both original memory and the derived memory by the learned transition model. The method to generate derived memory can be summarized as Equation (2), for the $i$-th derived trajectory, the deriving process contains two key steps: derived memory generation and trajectory imagination.

Derived memory generation:
$$h_t^i \sim \mathcal{N}(h_t, \sigma_h^2), \qquad s_t^i \sim \mathcal{N}(\mu_s, (\sigma_s + \sigma_{s'})^2)$$
Trajectory imagination:
$$h_t^i, s_t^i \sim p(h_t^i, s_t^i | h_{t-1}^i, s_{t-1}^i, a_{t-1}^i), \qquad a_t^i \sim \pi(a_t^i | s_t^i, h_t^i) \tag{2}$$

In order to ensure the reliability of the imagination with derived memory and further mitigate the model error's impact, we consider the reliability of the imagined trajectories by adding a trajectory evaluator. Every imagined trajectory is used with a reliability multiplier given by the evaluator. In the region where the model has reliable predict accuracy, the derived memory should be able to improve

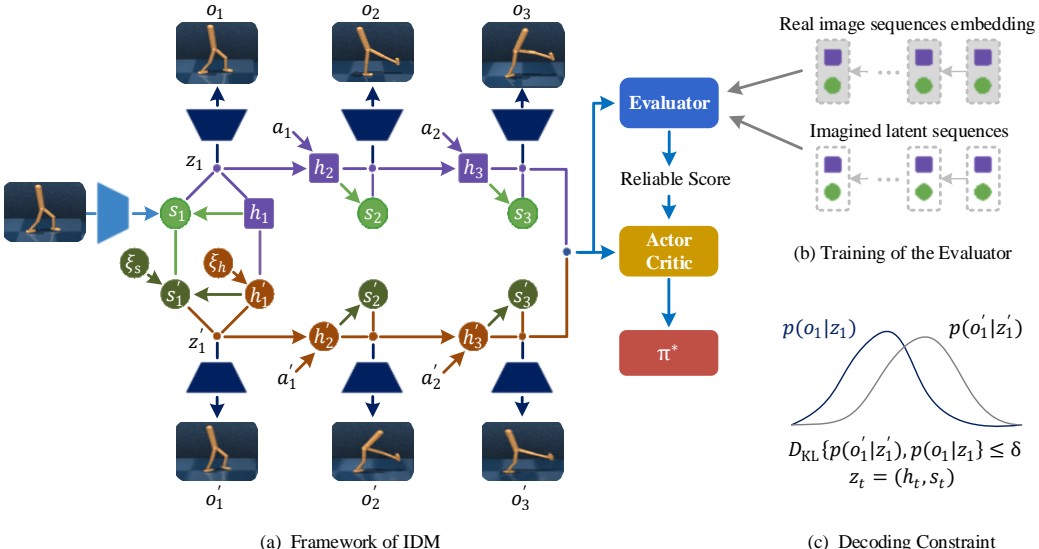

Figure 2: The overall framework of the IDM. The original memory $z_1 = (h_1, s_1)$ is embedded from a high-dimensional state, and the derived memory $(h'_1, s'_1)$ is generated by the transformation from the original memory. The decoding constraint is used to avoid reconstructed distortion. Latent trajectories are imagined from both the original memory and the derived memory. Each trajectory is given a reliability weight by the evaluator. The optimal policy $\pi^*$ is learned by the enriched imagination with prediction reliability weight under the actor-critic framework.

the policy robustness and sample efficiency significantly. We penalize the weight of the trajectories in the region with poor model prediction. The trajectory evaluator $\mathcal{E}$ aims to discriminate between real and overshoot sequences predicted by the model. In every iteration, we sample a number of state-action sequences generated by real interactions as real sequences $Seq_{\text{real}} = [z_\tau, a_\tau, z_{\tau+1}]_{\tau=t}^{t+T}$, use the latent model to predict the overshoot trajectories from the same start points as the fake sequences $Seq_{\text{fake}} = [z_t, a_t, \hat{z}_{t+1}] \cup [\hat{z}_\tau, a_\tau, \hat{z}_{\tau+1}]_{\tau=t+1}^{t+T}$. Train the evaluator to output the probability of whether a real trajectory, which is equal to maximize the following objective function

$$J_{eva} = \mathrm{E}_{(z_\tau, a_\tau) \sim D} \left\{ \sum_{\tau=t}^{t+N} \left[ \log \left( \mathcal{E} \left( z_\tau, a_\tau, z_{\tau+1} \right) \right) + \log \left( 1 - \mathcal{E} \left( \hat{z}_\tau, a_\tau, \hat{z}_{\tau+1} \right) \right) \right] \right\} \qquad (3)$$

where $\mathcal{E}(\cdot)$ is approximated by the combination of the MLP and the outermost Softmax and $z_t = \hat{z}_t$ since $z_t$ is the start point of the overshoot trajectory.

### 4.2 Policy Optimization

The policy $\pi_\varphi(z_\tau)$ is optimized by the imagined trajectories $\{z_\tau, a_\tau, r_\tau\}_{\tau=t}^{t+H}$. We estimate every state's value in the imagined trajectory to trade off the difficulty of long-horizon back-propagation and efficiency of gradient utilization. That means a single trajectory $\{z_\tau, a_\tau, r_\tau\}_{\tau=t}^{t+H}$ generates many sub-trajectories with different rollout steps. Every sub-trajectory is used to update the actor and critic with different reliability weight $w(z_\tau^i)$ given by the trajectory evaluator and is normalized in batch.

$$W(z_\tau^i) = \mathcal{E}(z_\tau^i, a_\tau^i, z_{\tau+1}^i), \quad z_\tau^i \sim (D^{ori}, D^{deri})$$

$$w(z_\tau^i) = \frac{W(z_\tau^i)}{\sum_{i=0}^{N} \sum_{\tau=t}^{t+H} W(z_\tau^i)} \qquad (4)$$

where $N$ is the batch size, $D^{ori}$ represents the trajectories imagined with original memory, and $D^{deri}$ represents the trajectories imagined with derived memory.

The value model $v_\psi(z_\tau)$ estimates the expected imagined rewards that the action model achieves from each state $z_\tau$. In an attempt to trade off bias and variance, we calculate the target state value by

TD-$\lambda$ method [21] as in Dreamer, which is an exponentially-weighted average of the estimates for different $k$ to make the estimation robust to different prediction horizons.

$$V_\lambda (z_\tau) \doteq (1 - \lambda) \sum_{n=1}^{H-1} \lambda^{n-1} V_N^n (z_\tau) + \lambda^{H-1} V_N^H (z_\tau) \tag{5}$$

$$V_N^k (z_\tau) \doteq E_{q_\varphi, q_\psi} \left( \sum_{n=\tau}^{h-1} \gamma^{n-\tau} r_n + \gamma^{h-\tau} v_\psi (z_h) \right)$$
$$h = \min(\tau + k, t + H) \tag{6}$$

where the $V_N^K$ estimates rewards beyond $k$ steps with the learned value model.

The critic is updated to regress the targets around which we stop the gradient [21]. The reliability weight is used in the critic updating, which is called weighted regression of target value. The objective for the critic $v_\psi (z_\tau)$, in turn, is to regress the target value

$$\min_\psi \left( \sum_{i=0}^{N} \left( \sum_{\tau=t}^{t+H} \frac{w(z_\tau^i)}{2} \| v_\psi (z_\tau^i) - V_\lambda (z_\tau^i) \|^2 \right) \right) \tag{7}$$

Different from DDPG [22] and SAC [23] which only maximize immediate Q-values, we leverage gradients through transitions and update the policy by backpropagation through the value model.

The objective of the actor with policy $\pi_\varphi(z_\tau)$ is to

$$\max_\varphi \left( \sum_{i=0}^{N} \left( \sum_{\tau=t}^{t+H} w(z_\tau^i) V_\lambda (z_\tau^i) \right) \right) \tag{8}$$

Since all steps are implemented as neural networks, we analytically compute $\nabla_\varphi J_{actor}$ by stochastic backpropagation. We use reparameterization for continuous actions and latent states,

$$a_\tau = \tanh \left( \mu_\varphi (s_\tau) + \sigma_\varphi (s_\tau) \epsilon \right), \quad \epsilon \sim \mathcal{N}(0, \mathbb{I}). \tag{9}$$

### 4.3 Theoretical analysis of IDM Framework

We give the upper bound of the value estimation error for MBRL under the IDM framework based on [24]. Denote the policy distribution shift between the current policy $\pi$ and the data-collecting policy $\pi_D$ as $\max_s D_{TV} (\pi \| \pi_D) \leq \epsilon_\pi$ by the maximum total-variation distance and the model generalization error as $\epsilon_m = \max E_{s \sim D} [D_{TV} (p(z', r \mid z, a) \| p'(z', r \mid z, a))]$, where $p(z', r \mid z, a)$ is the real state transition distribution and $p'(z', r \mid z, a)$ is the estimated state transition distribution. Under the IDM framework, if the model error under the updated policy is bounded by $\max E_{z \sim (D^{ori}, D^{aug})} [D_{TV} (p(z', r \mid z, a) \| p'(z', r \mid z, a))] \leq \epsilon_{m'}$ and the reweighting coefficient $w(z)$ given by the evaluator is bounded by $\max \{|w(z) - p(z', r \mid z, a) / p_\theta ((z', r \mid z, a)|\} \leq \epsilon_w$, then the returns estimation error upper bound is

$$\eta^{\text{branch}} [\pi] - \eta[\pi] \leq 2 r_{\max} \left[ \frac{\gamma^{k+1} \epsilon_\pi}{(1-\gamma)^2} + \frac{\gamma^k \epsilon_\pi}{(1-\gamma)} + \frac{k}{1-\gamma} \epsilon_w' \right] \tag{10}$$

where $\epsilon_w' = \epsilon_w + \epsilon_\pi \frac{d\epsilon_{w'}}{d\epsilon_\pi}$. The detailed proof can be seen in the Appendix A.5. $\epsilon_w$ indicates how well is the IDM helps to approximate the imagined distribution $p'(z', r \mid z, a)$ to the real distribution $p(z', r \mid z, a)$. The trajectories imagined with the derived memory increase the sample diversity, while the evaluator plays the role of importance sampling by reweighting the trajectories by $w(z)$. Coherent work of these two modules should effectively decrease the $\epsilon_w$, thus reducing the estimation error of the value function. If the evaluator is removed, the imagined trajectories, which are unreliable, will negatively impact the policy learning. If the derived memory module is removed, the effect of importance sampling will be weakened due to the lack of sample diversity. In addition, the positive effect on sample diversity of derived memory needs to be guaranteed by the decoding constraint in the R2S2 model, which is used to ensure the relevance between the derived memory and real physical states. Corresponding experiments are in Section 5.2.

Table 1: Performance comparison with image uncertainty

| | IDM | | Dreamer | | DrQ | |
|---|---|---|---|---|---|---|
| | 100 K | 500 K | 100 K | 500 K | 100 K | 500 K |
| Walker-Walk | $\mathbf{527.1} \pm 166$ | $\mathbf{914.9} \pm 61$ | $372.5 \pm 19$ | $876.3 \pm 48$ | $122.5 \pm 15$ | $151.9 \pm 23$ |
| Walker-Run | $\mathbf{159.2} \pm 32$ | $\mathbf{529.3} \pm 43$ | $116.1 \pm 11$ | $464.2 \pm 65$ | $58.3 \pm 14$ | $105.2 \pm 25$ |
| Hopper-Stand | $\mathbf{108.5} \pm 37$ | $\mathbf{841.2} \pm 54$ | $85.6 \pm 52$ | $430.2 \pm 128$ | $65.7 \pm 24$ | $252.6 \pm 38$ |
| Cartpole-Swingup | $\mathbf{329.6} \pm 58$ | $\mathbf{659.4} \pm 136$ | $303.3 \pm 43$ | $423.2 \pm 66$ | $112.6 \pm 17$ | $157.3 \pm 23$ |
| Cheetah-Run | $\mathbf{197.4} \pm 69$ | $\mathbf{598.2} \pm 42$ | $174.9 \pm 132$ | $401.34 \pm 104$ | $85.9 \pm 22$ | $96.3 \pm 38$ |
| Finger-Spin | $\mathbf{415.8} \pm 54$ | $\mathbf{505.5} \pm 61$ | $283.5 \pm 42$ | $374.9 \pm 88$ | $145.9 \pm 33$ | $436.3 \pm 27$ |

Table 2: Performance comparison without external image uncertainty (under the same setting in Dreamer)

| | IDM | | Dreamer | | DrQ | |
|---|---|---|---|---|---|---|
| | 100 K | 500 K | 100 K | 500 K | 100 K | 500 K |
| Walker-Walk | $\mathbf{630} \pm 113$ | $\mathbf{959} \pm 22$ | $277 \pm 12$ | $897 \pm 49$ | $612 \pm 164$ | $921 \pm 45$ |
| Walker-Run | $\mathbf{211} \pm 162$ | $\mathbf{536} \pm 25$ | $161 \pm 112$ | $482 \pm 68$ | $185 \pm 148$ | $459 \pm 139$ |
| Hopper-Stand | $\mathbf{216} \pm 73$ | $\mathbf{819} \pm 27$ | $94 \pm 48$ | $734 \pm 136$ | $106 \pm 51$ | $789 \pm 38$ |
| Cartpole-Swingup | $594 \pm 63$ | $824 \pm 63$ | $326 \pm 27$ | $762 \pm 27$ | $\mathbf{759} \pm 92$ | $\mathbf{868} \pm 10$ |
| Cheetah-Run | $328 \pm 48$ | $631 \pm 103$ | $235 \pm 137$ | $570 \pm 253$ | $\mathbf{344} \pm 67$ | $\mathbf{660} \pm 96$ |
| Finger-Spin | $623 \pm 76$ | $875 \pm 89$ | $341 \pm 70$ | $769 \pm 183$ | $\mathbf{901} \pm 104$ | $\mathbf{938} \pm 103$ |

# 5 Experiments

The IDM framework is implemented as Algorithm 1 (see Appendix A.2). We compare with both the SOTA of model-based methods and model-free methods. We mainly choose Dreamer as the model-based baseline to test the effectiveness of the IDM framework. As for model-free methods, DrQ [25] uses data regularized Q-learning method to learn the policy and shows a significant advantage in sample efficiency compared to CURL [26]. We choose the DrQ [25] as the model-free baseline.

## 5.1 Overall Performance

**Robustness to Uncertainty**: In an attempt to test the agent's ability to handle environmental uncertainty, the experiments are implemented in DMControl tasks [18] with observation uncertainty. Robotics may encounter observation uncertainties like the random swing of the input image caused by the robot's motion when the camera's fixed device is loose. We add a random rotation $(-5°, +5°)$ to the input image from the Mujoco environment (also used in RAD and DrQ paper as an augmented method) to simulate the possible random swing that may be experienced by robotics. Random rotation is a representative uncertainty the agent may encounter in the real world. During the training process of RL, images are only collected horizontally from the simulator, but in the real world, the camera could be skewed in a different direction, which changes the angles that agents or objects appear in the input image. Under this setting, we test the performance of IDM together with the Dreamer and the DrQ. The image uncertainty is added only at test time, and the purpose of this experiment is to test the robustness of policies learned by different methods when encountering unknown image uncertainties. The effectiveness of the IDM framework to tackle the input uncertainty is demonstrated in Table 1 and as Figure 3.

We also test the robustness of DrQ. Since DrQ's input is a stack of three sequential images (3 continuous images), we implement 2 different experiments: 1) randomly rotate the 3 images by the same angle and 2) randomly rotate the 3 images by different angles. All the rotate angles are sampled from $(-5°, 5°)$. The results show that under the first type of uncertainty DrQ's performance dropped significantly and under the second type of uncertainty DrQ almost failed. As shown in Figure 4, the training curve (the orange one) of DrQ demonstrates that it learns behaviour successfully; however, its performance decreases significantly when it is tested with image uncertainty, as illustrated in the testing curve (the blue one). The reason why DrQ performs poorly under external uncertainty is that DrQ has no guarantee to resist the uncertainty out of optimality invariant image transformations proposed in their paper, which especially constrains shifting under $\pm 4$ pixels from the original image.

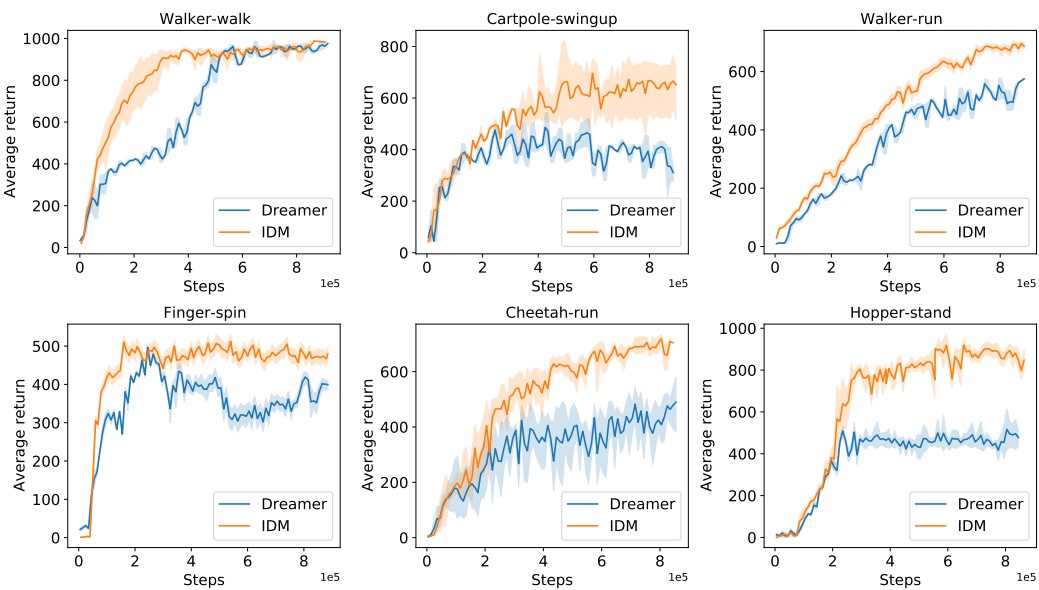

Figure 3: IDM exceeds Dreamer and DrQ at visual control tasks that testing with image uncertainty (the image uncertainty is added only at test time).

Our experiments show that although DrQ can effectively learn policy in training process with the given 5 random seeds, its performance on overcoming uncertainty is much worse than IDM.

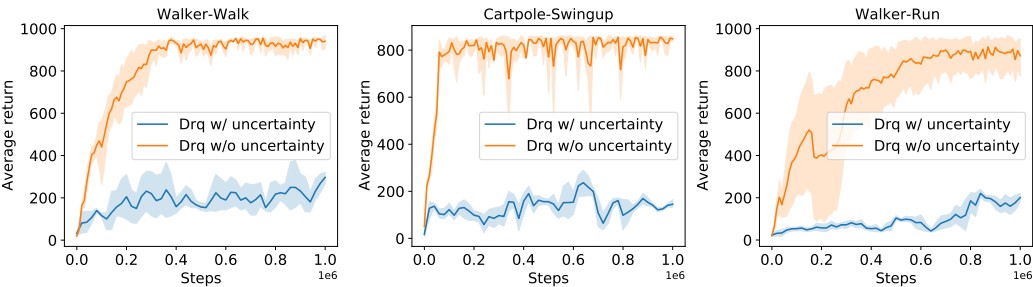

Figure 4: DrQ's performance decreased due to the observation uncertainty (the uncertainty is performed by the same parameter for each frame in the stack).

**Sample efficiency**: In order to show the sample efficiency of IDM, compared to Dreamer and DrQ [25], we test the performance of IDM in DMControl environments without image uncertainty with 5 random seeds. As illustrated in Table 2, IDM outperforms Dreamer and is comparable with the DrQ in environments without external uncertainty.

Our selection of the environment in Table 1 and Table 2 follows the below principle to ensure fairness: 3 tasks in which MBRL is more competitive than MFRL and 3 tasks in which MFRL is more competitive than MBRL. The measure of competitiveness is the comparison between the final performance of Dreamer and D4PG (top model-free method) with 1e9 steps shown in Dreamer paper.

Overall, the experimental results suggest that IDM outperforms Dreamer and DrQ in terms of robustness to uncertainty and further improves the sample efficiency of the model-based method, which proves that our proposed IDM can facilitate learning better behaviour with limited experiences.

## 5.2 Ablation Studies

In IDM, there are three key components: decoding constraint, derived memory and trajectory evaluator. We implement 4 groups of ablation experiments which are tested in DMControl tasks with

observation uncertainty. Note that the derived memory is based on the decoding constraint. If we keep the derived memory module without adding decoding constraint, it is theoretically invalid.

(1) **IDM without the evaluator(w/o EVA)**: In this setting, latent trajectories are imagined with derived memory but without the trajectory evaluator. Although it performs better than Dreamer in Walker-Walk, it has lower performance in Hopper-Stand due to lack of reliable guarantee for the derived data.

(2) **IDM without derived memory (w/o DM)**: In this setting, the constraint for RSSM and the trajectory evaluator are kept. As illustrated in Figure 5, it performs better than Dreamer but is insufficient compared with IDM. For instance, at the 100k steps of Walker-walk, its performance is 26.6% better than Dreamer, but IDM is 41.6% better than Dreamer.

(3) **IDM without decoding constraint and derived memory (w/o CST and DM)** : In this setting, only the trajectory evaluator is kept, which is used to avoid the damage caused by the trajectory with low prediction accuracy. The experimental results show that its performance is basically similar to Dreamer, which means that if we only use it to reweight the latent trajectories imagined with original memory, it makes little difference.

(4) **Dreamer with doubled batch size**: We compare IDM with Dreamer with two times of batch size than IDM (Doubled-Dreamer), as shown in Figure 5, Doubled-Dreamer shows less improvement than Dreamer, but IDM also outperforms it.

Overall, all the components in IDM are critical, and they complement each other. The importance of different components depends on the characteristic of the environment. For example, in Walker-Walk, which is a relatively stable bipedal walking environment compared to a single-legged robot, the generated derived memory does not hugely diverge from the real states experienced by the agent. Therefore, as shown in Figure 5(a) the derived memory provides a major contribution. On the contrary, due to the single-legged robot's instability, the derived memory generated by Hopper-stand usually contains poor quality states. Thus, as shown in Figure 5(c), the evaluator plays a significant role in this environment, but not in Walker. The ablation results in some difficult tasks, such as Walker-run (the agent needs to stand up first and then accelerate to run), the generated derived memory is not as helpful to improve the performance as the other environments, so does the evaluator. As shown in Figure 5(b), there is little difference in the results of ablation. This result also enlightens us to find a better way to construct derived memory in future work.

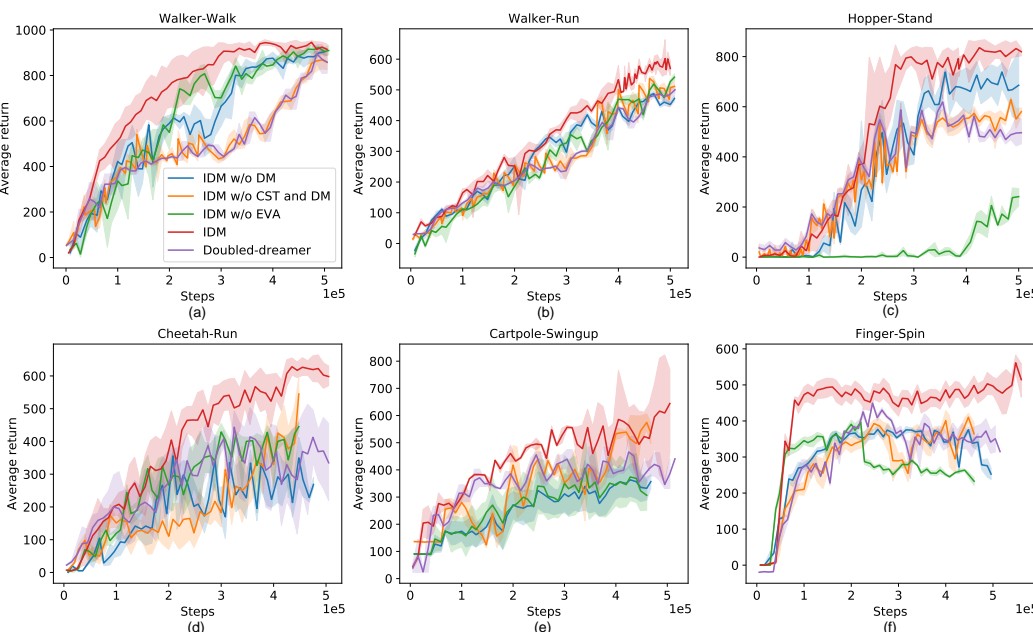

Figure 5: Contribution of each component in IDM to the performance improvement.

# 6 Related Works

(1) **Model-based policy optimization:** Prior works can be divided into the following categories, including Dyna-like algorithms, value expansion, backpropagation through paths, and sampling-based planning. Dyna-like algorithms alternate between model learning from environmental interaction, data generation, and policy improvement by model-free methods [5], e.g., ME-TRPO [6]. Value expansion algorithm utilizes the dynamic model to improve the estimation accuracy of the cumulative return [11]. The backpropagation through path method is built upon using the analytic gradient of state value backpropagate through the dynamic transition. Typical algorithms in this category include Dreamer [4], PILCO [27], and SVG [14]. Dreamer is the most representative model-based algorithm for visual control tasks. The idea of sampling-based planning is to choose the best action by a large number of samples and regard it as the objective for the policy network. Representative algorithms include the cross-entropy method [28] in continuous space, which is used in PlaNet [8] and PETS [9], and MCTS [10] in discrete space. MuZero further incorporates MCTS and achieves remarkable performance on Atari, and board games [29].

(2) **Latent world model:** In MBRL for high dimension input tasks, the latent world model offers a flexible way to represent key information of the observations. World Models [1] learn latent dynamics in a two-stage process: representation learning and latent dynamics learning. However, it lacks the coordination of the two processes. PlaNet [2] proposes a recurrent stochastic state model (RSSM), which learns them jointly. Dreamer [4] utilizes the RSSM to make the long-term imagination. [30] represents the model uncertainty by neural network ensemble to approximately infer model posteriors and further improves the performance of PlaNet. DreamerV2 [31] proposes discrete world models for discrete visual control tasks like Atari games. BIRD [3] improves the prediction accuracy by maximizing the mutual information between the trajectories imagined with memory and real trajectories but does not consider the diversity of memory. Recently, contrastive learning-based approaches are gathering increasing attention. It has been applied to learn latent world models [4, 32, 33, 34] motivated from different perspectives. Hafner et al.[4] try to use contrastive learning as an alternative to image reconstruction, however, the contrastive learned agent gives worse performance compared with the one learned by image reconstruction. Dreaming [33] derives an InfoMax objective of contrastive learning from the evidence lower bound of Dreamer and removes the decoder to solve the object vanishing problem of the representation methods based on image reconstruction. CVRL [34] emphasizes the strength of contrastive learning in handling complex visual observations and uses a hybrid training scheme that includes a model-based scheme (Dreamer) and a model-free scheme (SAC). Since the key idea of IDM is not in conflict with contrastive learning, we can extend our framework by incorporating advanced contrastive representation learning to further improve the overall performance, which is left as future work.

(3) **Data augmentation for reinforcement learning:** Data augmentation has been investigated in the context of RL to improve generalization and sample efficiency. CURL [26] minimize the contrastive loss between an image and its augmented version to learn a better latent representation. RAD [35] attempts to directly train the RL objective on multiple augmented views of the observations without any auxiliary loss. DrQ [25] aims to solve the distribution mismatch problem in off-policy RL, which utilized random cropping and regularized Q-functions in conjunction.

# 7 Conclusion

We present a novel model-based framework, called **I**magining with **D**erived **M**emory (IDM), aiming to improve the policy robustness and the sample efficiency by effectively enhancing the agent's imagination diversity. IDM optimizes a parameterized policy with the extended imagined trajectories from both original and derived memory by propagating analytic gradients of multi-step values back through learned latent dynamics. It outperforms previous methods in terms of the robustness to uncertainty and further improves the sample efficiency of the model-based method on a variety of challenging continuous control tasks with image inputs. Other methods to generate derived memory, such as employing different assumptions for latent noise distribution, will be investigated in the future. Furthermore, since our method is compatible with representation learning techniques, extending our framework by incorporating advanced contrastive representation learning (e.g., RAD, CURL, and Dreaming ) to further improve the sample efficiency could also be left as future work.

## Broader Impact

Sample efficiency and policy robustness are key factors for the development of real-world applications on automatic control, such as robotics and autonomous driving. The proposed IDM algorithm is situated in model-based RL and learns behaviours via imagination with derived memory, which enables the agent to tackle unseen situations by efficient enhancement on imagination. Thus it further improves policy robustness and sample efficiency over existing works, which provides a broader prospect for real-world applications. The technique to generate and utilize the derived memory is wide applicable computational efficient and could motivate both research and the industrial community to refine the presented method for addressing the uncertainty in different sources of collected data and making up the gap between the real world and training environments. In the long run, this paper contributes to the foundation for realizing robust artificial intelligence, which will improve efficiency, reduce cost and risk in industrial operation, agriculture and also facilitate people's life by enhancing daily life intelligence levels.

## Acknowledgments and Disclosure of Funding

The authors would like to thank the anonymous reviewers for their valuable comments and helpful suggestions. The work is supported by Huawei Noah's Ark Lab; Ping Luo is supported by the General Research Fund of Hong Kong No.27208720.

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
