# Model-Based Reinforcement Learning via Imagination with Derived Memory

**Yao Mu**
The University of Hong Kong
muyao@connect.hku.hk

**Yuzheng Zhuang** *
Huawei Noah's Ark Lab
zhuangyuzheng@huawei.com

**Bin Wang**
Huawei Noah's Ark Lab
wangbin158@huawei.com

**Guangxiang Zhu**
Tsinghua University
guangxiangzhu@outlook.com

**Wulong Liu**
Huawei Noah's Ark Lab
liuwulong@huawei.com

**Jianyu Chen**
Tsinghua University
jianyuchen@tsinghua.edu.cn

**Ping Luo**
The University of Hong Kong
pluo@cs.hku.hk

**Shengbo Eben Li**
Tsinghua University
lishbo@tsinghua.edu.cn

**Chongjie Zhang**
Tsinghua University
chongjie@tsinghua.edu.cn

**Jianye Hao**
Huawei Noah's Ark Lab
haojianye@huawei.com

## A    Appendix

### A.1    Detailed Structure of R2S2 Model

The robust world model consists of the representation model, the transition model, the observation model, the reward model and a decoding constraint which is used to ensure the relevancy between the derived memory and real physical states.

The representation model aims to infer approximate state posteriors from past observations and actions, where $q\left(s_t \mid z_{t-1}, a_{t-1}, o_t\right)$ is a diagonal Gaussian with mean and variance parameterized by a convolutional neural network (CNN) followed by a fully connected neural (FC) network. In order to enable accurate long-term predictions, the transition model is designed with both stochastic and deterministic paths. The latent state is split into a stochastic state $s_t$ and a deterministic hidden state $h_t$, where the stochastic state $s_t$ is Gaussian with mean and variance parameterized by a fully connected neural network. The transition model $f\left(h_{t-1}, s_{t-1}, a_{t-1}\right)$ is implemented as a recurrent neural network (RNN). The observation model is Gaussian with mean parameterized by a transposed convolutional neural network and identity covariance. The reward model $q(r_t|z_t)$ is a scalar Gaussian with mean parameterized by a fully connected (FC) neural network and unit variance.

To improve the latent model's robustness, we add a decoding constraint to the model learning, which ensures the consistency between derived latent states and the real physical states. We perform random transformation on the latent state, in which the noise $\xi_h$ is added to the hidden state $h_t$ and the noise $\xi_s$ is added to $s_t$. The decoding constraint is between the $o_t$ which is reconstructed from $(h_t, s_t)$ and the $o'_t$ which is reconstructed from $(h'_t, s'_t)$.

---

*Yuzheng Zhuang is the corresponding author. Yao Mu conducted this work during the internship in Huawei Noah's Ark Lab.

35th Conference on Neural Information Processing Systems (NeurIPS 2021).

## A.2 Algorithm Flowchart

The IDM framework is implemented as Algorithm 1, which mainly contains 4 key processes: (1) learning the R2S2 model, (2) imagination from derived memory and original memory, (3) behaviour learning, and (4) environment interaction. We denote imagined quantities with $\tau$ as the time index. You can see details from the Algorithm 1.

---

**Algorithm 1:** Imagination with Derived Memory for MBRL (IDM)

---

Initialize dataset $D$ with $S$ random seed episodes.
Initialize neural network parameters $\theta$, $\varphi$ and $\psi$.
**while** *not convergence* **do**

    **for** *update step* $c = 1 \dots C$ **do**

        // Learning the R2S2 model

        Draw $\mathcal{B}$ data sequences $\{(a_t, o_t, r_t)\}_{t=k}^{k+L} \sim \mathcal{D}$.

        Compute latent state $z_t \sim p_\theta (z_t \mid z_{t-1}, a_{t-1}, o_t)$.

        Update $\theta$ by $\min_\theta \{J_M\}$ with the decoding constraint.

        //Imagination with derived memory and original memory

        Derived memory generation: $z_t' = (h_t', s_t')$, $h_t' \sim N(h_t, \sigma_h^2)$, $s_t' \sim N(\mu_s, (\sigma_s + \sigma_{s'})^2)$.

        Imagine trajectories $\{(z_\tau, a_\tau)\}_{\tau=t}^{t+H}$ from each $z_t$ and $\{(z_\tau', a_\tau')\}_{\tau=t}^{t+H}$ from each $z_t'$.

        Compute the reliability weights $w(z_\tau)$ and $w(z_\tau')$ by the evaluator and normalize them in the mini batch.

        Predict rewards $\mathrm{E}\left(q_\theta \left(r_\tau \mid z_\tau\right)\right)$ and values $v_\psi \left(z_\tau\right)$.

        Compute value estimates $\mathrm{V}_\lambda \left(z_\tau\right)$.

        // behaviour learning

        Update $\varphi \leftarrow \varphi + \alpha \nabla_\varphi \sum_{\tau=t}^{t+H} w(z_\tau) \mathrm{V}_\lambda \left(z_\tau\right)$.

        Update $\psi \leftarrow \psi - \alpha \nabla_\psi \sum_{\tau=t}^{t+H} \frac{w(z_\tau)}{2} \left\| v_\psi \left(z_\tau\right) - \mathrm{V}_\lambda \left(z_\tau\right) \right\|^2$.

    // Environment interaction

    $o_1 \leftarrow$ env.reset () .

    **for** *time step* $t = 1 \dots T$ **do**

        $r_t, o_{t+1} \leftarrow$ env.step $(a_t)$.

    Add experience to dataset $\mathcal{D} \leftarrow \mathcal{D} \cup \left\{(o_t, a_t, r_t)_{t=1}^T\right\}$.

---

## A.3 Video Prediction by the R2S2 Model

As shown in Figure 1, our R2S2 model achieves pixel-accurate predictions in various environments. We randomly selected action sequences from test episodes collected with action noise alongside the training episodes. The first row is the ground-truth image input, the second row is the image reconstructed with the random noise on the latent states, and the third row is reconstruction error.

## A.4 Experimental details and Hyper Parameters

We use a single Nvidia K80 GPU for each training run. We implement our algorithms based on the Tensorflow-v2 version code [2] of Dreamer released by its author. We use the same hyperparameters of Dreamer across all continuous control tasks. The main parameters of the IDM algorithm are listed in Table 1. The detailed training process of the trajectory evaluator is shown in Figure 2.

Here, we discuss some tricks used in the implementation. We scale down gradient norms that exceed 100 and clip the KL-divergence in Equation (1) below 3 free nats as in Dreamer and PlaNet. We compute the $v_\lambda$ targets with $\gamma = 0.99$ and $\lambda = 0.95$. The dataset is initialized with $S = 5$ episodes collected using random actions. We iterate between 100 training steps and collect 1000 new transition data. At the beginning of the algorithm, we add a zero-mean Gaussian noise with a standard deviation of 5 to the initial policy when collecting data. Instead of manually selecting the action repeat for each environment as in DrQ and Rad, we fix the action repeat to 2 for all environments like Dreamer. The DrQ algorithm used in the comparative experiments is implemented with official open-source code [3].

---

[2]https://github.com/danijar/dreamer
[3]https://github.com/denisyarats/drq

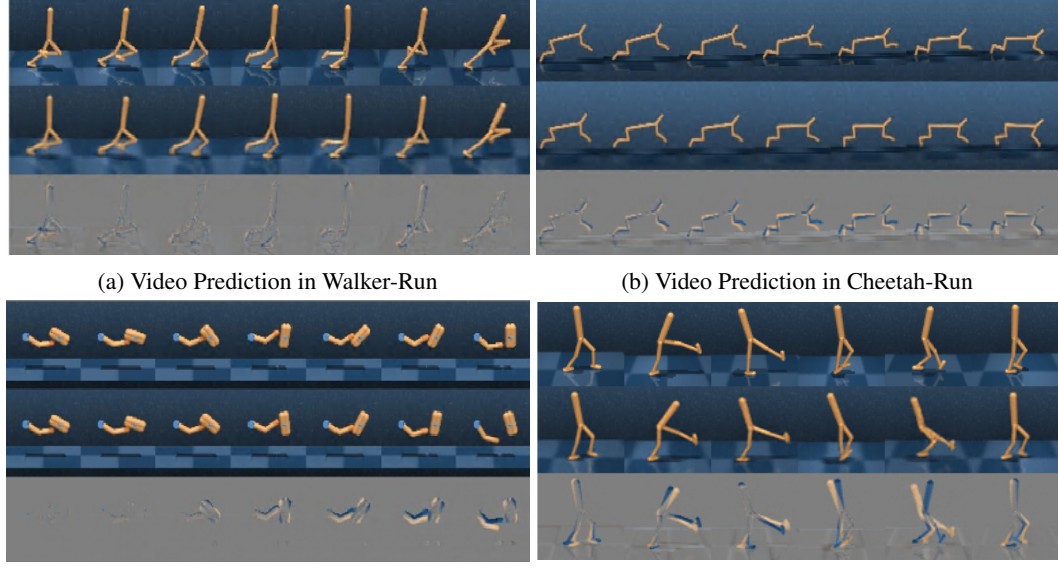

(a) Video Prediction in Walker-Run

(b) Video Prediction in Cheetah-Run

(c) Video Prediction in Finger-Spin

(d) Video Prediction in Walker-Walk

Figure 1: Video Prediction by the R2S2 Model

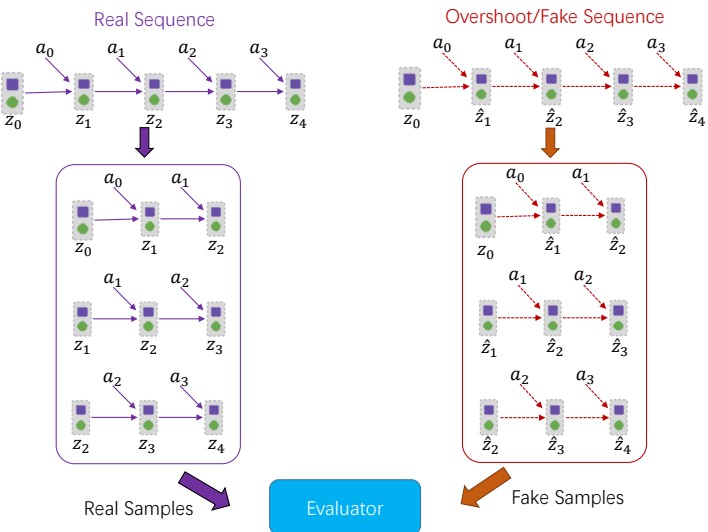

Figure 2: The training process of the evaluator

## A.5 Proof for Returns Estimation Error Upper Bound

In this section, we prove the upper bound of the value estimation error for MBRL under the IDM framework based on Janner's work [1]. The gap between true returns and model returns can be expressed in terms of two error quantities of the model: generalization error $\epsilon_m$ due to sampling, and distribution shift $\epsilon_\pi$ due to the updated policy encountering states not seen during model training.

We denote the policy distribution shift between the current policy $\pi$ and the data-collecting policy policy as $\max_z D_{TV}(\pi \| \pi_D) \le \epsilon_\pi$ by the maximum total-variation distance and the model generalization error as $\epsilon_m = \max E_{z \sim D}[D_{TV}(p(z', r \mid z, a) \| p'(z', r \mid z, a))]$.

Next, we analyze the IDM framework based on Janner's work [1]. Denote $p_\theta(z' \mid z, a)$ as the state transition probability predicted by model. The data which used to optimize the policy in

Table 1: Hyper parameters table

| Name | Symbol | Value |
|---|---|---|
| Collect interval | $C$ | 100 |
| Interact interval | $T$ | 1000 |
| Batch size | $\mathcal{B}$ | 50 |
| Size of $s$ | $n_s$ | 30 |
| Size of $h$ | $n_h$ | 200 |
| Sequence length | $L$ | 50 |
| Imagination horizon | $H$ | 15 |
| Learning rate for model | $\alpha_0$ | $6e-4$ |
| Learning rate for critic | $\alpha_1$ | $8e-5$ |
| Learning rate for actor | $\alpha_2$ | $8e-5$ |
| Scale of the KL divergence | $\beta$ | 1 |
| Constraint penalty coefficient | $\beta'$ | 1 |

the IDM framework is reweighted by $w(z)$. Therefore, the $p'\left(z' \mid z, a\right)$ in the IDM is equal to $w(z)p_\theta\left(z' \mid z, a\right)$. In the IDM framework, the reliability weight $w(z)$ aims to approximate the important sampling factor between the data from the real environment and the imagined data, and it is the most significant difference between IDM and Dreamer. We assumed that $w(z)$ is bounded by

$$\max\left\{\left|w(z) - \frac{p\left(z' \mid z, a\right)}{p_\theta\left(z' \mid z, a\right)}\right|\right\} \leq \epsilon_w. \tag{1}$$

Thus, the total variance distance between the $p_\theta\left(z' \mid z, a\right)$ and $p\left(z' \mid z, a\right)$ can be derived as

$$D_{TV}\left(p\left(z' \mid z, a\right) \| p'\left(z' \mid z, a\right)\right)$$
$$= \int_{(z,a,z')} \left|p\left(z' \mid z, a\right) - w(z)p_\theta\left(z' \mid z, a\right)\right|. \tag{2}$$

When $p\left(z' \mid z, a\right) - w(z)p_\theta\left(z' \mid z, a\right) > 0$, we have

$$\int_{(z,a,z')} p\left(z' \mid z, a\right) - w(z)p_\theta\left(z' \mid z, a\right)$$
$$= \int_{(z,a,z')} p\left(z' \mid z, a\right) - w(z)p_\theta\left(z' \mid z, a\right)$$
$$\leq \int_{(z,a,z')} p\left(z' \mid z, a\right) - \left(\frac{p\left(z' \mid z, a\right)}{p_\theta\left(z' \mid z, a\right)} - \epsilon_w\right)p_\theta\left(z' \mid z, a\right) \tag{3}$$
$$= \int_{(z,a,z')} \epsilon_w p_\theta\left(z' \mid z, a\right) = \epsilon_w.$$

When $p\left(z' \mid z, a\right) - w(z)p_\theta\left(z' \mid z, a\right) \leq 0$, we have

$$\int_{(z,a,z')} p\left(z' \mid z, a\right) - w(z)p_\theta\left(z' \mid z, a\right)$$
$$= \int_{(z,a,z')} w(z)p_\theta\left(z' \mid z, a\right) - p\left(z' \mid z, a\right) \tag{4}$$
$$\leq \int_{(z,a,z')} \left(\epsilon_w + \frac{p\left(z' \mid z, a\right)}{p\left(z' \mid z, a\right)}\right)p\left(z' \mid z, a\right) - p\left(z' \mid z, a\right) = \epsilon_w.$$

Therefore the $\epsilon_m$ could be bounded by $\epsilon_w$. The total variance distance between the $p'\left(z' \mid z, a\right)$ and $p\left(z' \mid z, a\right)$ can be derived as (5).

$$D_{TV}\left(p\left(z' \mid z, a\right) \| p'\left(z' \mid z, a\right)\right)$$
$$= \int_{(z,a,z')} \left|p\left(z' \mid z, a\right) - w(z)p_\theta\left(z' \mid z, a\right)\right| \leq \epsilon_w. \tag{5}$$

Thus, the $\epsilon_m$ is bounded by $\epsilon_w$. If we can instead approximate the model error on the distribution of the current policy $\pi$, which we denote as $\epsilon'_m$, and approximate the $\epsilon'_m$ with a linear function of the policy divergence yields: $\hat{\epsilon}_{m'}(\epsilon_\pi) \approx \epsilon_m + \epsilon_\pi \frac{\mathrm{d}\epsilon_{m'}}{\mathrm{d}\epsilon_\pi}$, the upper bound of returns estimation error of k-branched model rollout is illustrated as the Theorem A.5.1.

**Theorem A.5.1** . Under the IDM framework, if the model error under the updated policy is bounded by $\epsilon_{m'} \geq \max E_{z \sim (D^{ori}, D^{aug})} [D_{TV}(p(z' \mid z, a) \| p'(z' \mid z, a))]$ and the reweighting coefficient is bounded by $\max \{|w(z) - p(z' \mid z, a)/p_\theta(z' \mid z, a)|\} \leq \epsilon_w$, then the returns estimation error upper bound is

$$\eta^{\text{branch}}[\pi] - \eta[\pi] \leq 2r_{\max}\left[\frac{\gamma^{k+1}\epsilon_\pi}{(1-\gamma)^2} + \frac{\gamma^k\epsilon_\pi}{(1-\gamma)} + \frac{k}{1-\gamma}\epsilon'_w\right] \tag{6}$$

where $\epsilon'_w = \epsilon_w + \epsilon_\pi \frac{\mathrm{d}\epsilon_{w'}}{\mathrm{d}\epsilon_\pi}$.

**Proof:** As in the proof for Theorem A.5.1, the proof for this theorem requires adding and subtracting the correct reference quantity and applying the corresponding returns bound (the Lemma B.4 in [1]).

The choice of reference quantity is a branched rollout which executes the old policy $\pi_D$ under the true dynamics until the branch point, then executes the new policy $\pi$ under the true dynamics for $k$ steps. We denote the returns under this scheme as $\eta^{\pi_D, \pi}$. We can split the returns as follows:

$$\eta[\pi] - \eta^{\text{branch}} = \underbrace{\eta[\pi] - \eta^{\pi_D, \pi}}_{L_1} + \underbrace{\eta^{\pi_D, \pi} - \eta^{\text{branch}}}_{L_2}. \tag{7}$$

We can bound both terms $L_1$ and $L_2$ using the Lemma B.4 in [1]. $L_1$ accounts for the error from executing the old policy instead of the current policy. This term only suffers from error before the branch begins, and we can use the Lemma B.4 in [1], setting $\epsilon_\pi^{\text{pre}} \leq \epsilon_\pi$ and all other errors set to 0. This implies:

$$|\eta[\pi] - \eta^{\pi_D, \pi}| \leq 2r_{\max}\left[\frac{\gamma^{k+1}}{(1-\gamma)^2}\epsilon_\pi + \frac{\gamma^k}{1-\gamma}\epsilon_\pi\right]. \tag{8}$$

$L_2$ incorporates model error under the new policy incurred after the branch. Again we use the Lemma B.4 in [1], setting $\epsilon_m^{\text{post}} \leq \epsilon_m$ and all other errors set to 0. This implies:

$$|\eta[\pi] - \eta^{\pi_D, \pi}| \leq 2r_{\max}\left[\frac{k}{1-\gamma}\epsilon_{m'}\right] \leq 2r_{\max}\left[\frac{k}{1-\gamma}\epsilon_{w'}\right]. \tag{9}$$

Adding $L_1$ and $L_2$ together, the returns estimation error upper bound can be derived as

$$\eta^{\text{branch}}[\pi] - \eta[\pi] \leq 2r_{\max}\left[\frac{\gamma^{k+1}\epsilon_\pi}{(1-\gamma)^2} + \frac{\gamma^k\epsilon_\pi}{(1-\gamma)} + \frac{k}{1-\gamma}\epsilon_{w'}\right]. \tag{10}$$

## References

[1] Michael Janner, Justin Fu, Marvin Zhang, and Sergey Levine. When to trust your model: Model-based policy optimization. In *Advances in Neural Information Processing Systems*, pages 12519–12530, 2019.