# OpenReview forum: "Model-Based Reinforcement Learning via Imagination with Derived Memory"
_NeurIPS.cc/2021/Conference — NeurIPS 2021 Poster_

### Official Review · Reviewer_URjo · 2021-07-14

**Rating:** 6
**Confidence:** 4

**Summary:**

Summary

In this paper, the authors consider the latent space in imaginary space.
A GAN styled latent space noise is trained to add more diversity and robustness to imaginary planning.


**Ethical Concerns:**

no ethical concerns.

**Limitations And Societal Impact:**

limitation addressed and no obvious negative societal impact found in the paper.

**Main Review:**



Main Review

Pros:
1. The algorithm is neat and well explained.
Good related work is provided in the paper, and the algorithm is well explained.

2. Theoretical analysis is provided in the paper, making the algorithm more convincing and providing good intuition and inspiration to future research.

3. The problem studied is challenging and interesting.
Image space continuous control problem is considered one the hardest problems in RL or robotics. This paper will contribute to this line of research.

Cons:
1. The novelty is lacking.
The algorithm’s contribution is relatively incremental compared to the existing research,
given that it is built upon Dreamer and the elements that are novel is the added noise.
But still it remains unclear how robust or scalable the algorithm is (from the experiment section),
as shown in my later comments,
the experiments have a major flaw of not able to reproduce the performance of the original paper which open-sourced their code-base.
That being sad, it is hard to draw any conclusions of how good the proposed algorithm is in terms of practical performance. (updated according to AC's request)


2. The experiment section is not convincing.
Mostly I believe the only dreamer is used as the baseline.
And the performance of dreamer on different environments tested is having a large gap between the performance reported in the original paper.
For example cartpole-swingup is around 800 in the original dreamer paper, but was shown around 400 in the given paper. The case is consistent in Walker-Run, Finger-Spin, Cheetah-Run, Hopper-Stand.
Considering Dreamer is open-sourced, I wonder if there's a good explanation for the performance gap.


Updated after the rebuttal.

**Time Spent Reviewing:**

2 hours

---

> ### Author Response · Authors · 2021-08-08
> **Response to Reviewer URjo**
>
> Thanks for your comments, although some of them we wish to sincerely discuss with you.
> We notice that your major criticism is that we "fail" to reproduce the performance of Dreamer, and thus lead to the whole experimental results being less convincing to support our conclusions. Therefore, we first clarify your concerns about the experiments then we will reiterate the novelty of our proposed IDM as a solution to solve the problem that you also mentioned is worthy of studying.
>
>
> Q1: The experiments have a major flaw of not being able to reproduce the performance of the original paper, which open-sourced their code-base
>
> Response:
>
> First of all, we reproduce Dreamer's performance for all tasks you mentioned with the open-source code of Dreamer, including Cartpole-Swingup, Walker-Run, Finger-Spin, Cheetah-Run and Hopper-Stand and shown in Table 2, to support our conclusion that IDM has better sample efficiency than Dreamer. Actually, the reproduced performances in some of the tasks are even better than those given in Dreamer's paper, e.g.the cumulative reward of Cartpole-Swingup in 500k is 762 $\pm$ 27 as shown in our paper, which is around 800 in the original Dreamer paper.
>
> Furthermore, we compare the performance under the experiment setting with external input image uncertainty in Table 1, and Dreamer's performance drops while tackling inputs uncertainty, (for example, cartpole-swingup is around 400, as you mentioned). The overall results demonstrate IDM's advantages in robustness. Finally, Dreamer is not the only baseline. We also compare with DrQ, which is the state-of-the-art model-free method under both experiments settings, as shown in Table 1 and Table 2.
>
>
> Q2: The novelty is lacking, it is built upon Dreamer, and the elements that are novel is the added noise.
>
> Response:
>
> IDM is not simply adding noise in the latent state but novelly designs a complete framework for the generation and utilization of derived memory.
>
> The principal idea of our proposed approach is to generate diverse and reliable imagined rollouts from disturbances added latent states, called derived memory, to facilitate robustness and better sample efficiency. To ensure the benefit of the augmented imaginations, we novelly design a coherently learning mechanism for the constraint and the evaluator modules to stabilize the learning. Otherwise, it would lead to low-fidelity reconstruction (Figure 1) and performance damaging (Figure 5, green curves). Besides, our approach can be applied to most of the other model-based RL methods since the underlying track information is typically employed during policy learning.
>
> We find that you seem to miss most of the details of our experimental settings and results, thus we briefly clarify the whole picture of our proposed approach in this rebuttal session and hope this can resolve most of your concerns. We sincerely wish you to value our work and reconsider the score.

---

> > ### Comment · Reviewer_URjo · 2021-08-16
> > **Thank you for the experiment clarification**
> >
> > After reading the authors' rebuttal I realize I mistakenly thought the curves from Figure 3 as the normal curve comparison figures.
> > The rebuttal address my biggest concern and I am willing to increase my rating accordingly.
> > I would also like to suggest the authors to make a few curves from Table 1 and Table 2 and extend the number of samples to 1e6,
> > which can be very powerful figures since they can support the major claim in the paper.

---

> > > ### Author Response · Authors · 2021-08-17
> > > **Thanks for your responsible reply**
> > >
> > > Thanks for your responsible reply and conductive suggestion. We will extend the number of samples to 1e6 in the curve of Episode Return accordingly.

---

### Official Review · Reviewer_S88K · 2021-07-15

**Rating:** 6
**Confidence:** 4

**Summary:**

The authors introduce a variant of the model-based Dreamer algorithm with an added constraint at training time: that perturbed latents (with Gaussian noise) should lead to similar observation reconstructions. This is leveraged to augment the real trajectories with imagined trajectories that start from nearby latent states. Finally, a learned discriminator of real vs imagined latent trajectory is employed to reweight the trajectories in the policy optimization objective. The approach is tested in continuous control domains.


**Limitations And Societal Impact:**

Yes.

**Main Review:**

The approach proposed by the authors is reasonable and novel but the writing is unfortunately a bit hard to follow at times and the language is imprecise/inconsistent in places.

For example, it would make sense to be more precise about what constitutes real and derived memories (cf L121) given this is a crucial aspect of the paper. Sometimes this seems to be interchanged with real and “overshoot” (or fake) sequences or trajectories. The notation around these concepts doesn’t help:
- L140: Seq_fake is defined using $\hat\tau_{t+1}$, but it’s not clear what this is.
- On page 5, $z$ sometimes refers to a single latent, and sometimes seems to refer to a full trajectory (e.g. Eq 4).
- $\mathcal{E}$ takes a sequence as input in Eq3, and then is given a transition tuple as input in Eq 4.

The exposition in Alg 1 in the appendix does clarify some things, but the notation should be made consistent.

The experimental work compares the proposed approach (IDM) to Dreamer and DrQ on some Mujoco domains. In the normal conditions (Eq 2), IDM performs similarly to DrQ (if anything DrQ performs better since the difference between IDM and DrQ in the Walker tasks does not seem to be statistically significant). Nevertheless, the fact that it consistently outperforms the baseline (Dreamer) suggests that the mechanisms of the approach are beneficial in these domains. The provided ablations reinforce that point.

The experiments with the randomized visual inputs are a bit odd. IDM performs a lot better than Dreamer+DrQ there.
- Is the randomization done during the training as well or only at test time?
- It’s not clear that perturbing latents would match the image rotation perturbation? So why is IDM performing better here?

Overall, the approach may be promising but I don’t think the paper is polished enough in its current form to be accepted.

More comments / questions:
- The transition for the deterministic latent variables h is not explained in the paper.
- Fig 1: left/right should be the same top trajectories, otherwise it’s not really a fair comparison. For example the trajectory in a) shows a falling walker vs a standing walking in b), small perturbations in these regions might have completely different effects.
- Many parts of the paper repeat the equations from the Dreamer paper. In this case please say “as in Dreamer” rather than “similar to Dreamer” if the approach is unchanged (Eq 5,6). Similar comment for L160-161, this is not specific to your approach.
- The relation of the analysis in 4.3 to [31] is not mentioned.
- How was the noise variance for the latent perturbation chosen? Was the regular transition noise variance similarly tuned?

Minor:

- L162 object -> objective
- Sentence on L33 should be rephrased.
- Figure 3 caption doesn’t make sense.



**Time Spent Reviewing:**

4

---

> ### Author Response · Authors · 2021-08-08
> **Response to Reviewer S88K**
>
> Thanks for your constructive comments and helpful criticism, we sincerely appreciate your positive comments on our proposed approach as a novel and promising solution to solve the problem. And we notice that the major criticism is about the imprecise/inconsistency exists in our language, thus detailed responses regrading each problem are listed below.
>
> Q1: Is the random image uncertainty added during the training as well or only at test time?
>
> Response:
>
> The image uncertainty is added only at test time. The purpose of this experiment is to test the robustness of policies learned by different methods when encountering unknown image uncertainties.
>
> Q2: It’s not clear that perturbing latent would match the image rotation perturbation? So why is IDM performing better here?
>
> Response:
>
> Firstly the random rotation does not require the noise must obey Gaussian distribution, and other distributions like t-distribution are also appropriate in terms of the diversity of latent states. The reason for us adding Gaussian perturbation is that it is the most intuitive way since the stochastic part of latent $s$ in RSSM obeys the Gaussian distribution. Furthermore, the Gaussian noise in latent space could be theoretically mapped to any types of uncertainty in image level, and this principle is extensively used in VAE and GAN.
>
> Our method aims to cover the influence in latent space cause by any image uncertainty. By adding Gaussian noise to the latent state, the agent can learn behaviors in a larger latent space. The size of the derived latent space is determined by the variance of the  Gaussian noise and the variance of the original latent state. The input image with uncertainty (no matter which kind of uncertainty) will be encoded to latent space. If the encoded latent state is in the enlarged latent space, the policy can perform well since it could encounter this situation with the imagined trajectory with derived memory.
> Our method should be universal to different kinds of image uncertainty, and evaluation with other types of image uncertainty will be added to future work.
>
> Q3: The transition for the deterministic latent variables h is not explained in the paper.
>
> Response:
>
> Thanks for your kindly remind, we will add a detailed explanation of the transition for $h$ in our paper.
> The transition for the deterministic latent variables is the same as the method in Dreamer, which is updated by a recurrent model.  The $f(h_{t-1}, s_{t-1}, a_{t-1})$ is implemented as a recurrent neural network (RNN).
>
> Q4: Fig 1: left/right should be the same top trajectories, otherwise it’s not really a fair comparison.
>
> Response:
>
> Thanks for your great advice.
> In our experiments, we randomly select a trajectory from the replay buffer and add noise to the latent state at each time for testing, which is the same in the derived memory generation. Effective derived memory requires that a noise added latent state in any trajectory should correspond to a real physical state, i.e., ensure the reconstructed images are not distorted. The way you suggested is more prudent and precise for visualization, you can see the corresponding results on " https://sites.google.com/view/idm-mbrl ". We will modify this figure accordingly in our paper.
>
>
> Q5: Many parts of the paper repeat the equations from the Dreamer paper. In this case, please say "as in Dreamer" rather than "similar to Dreamer" if the approach is unchanged.
>
> Response:
>
> Thanks a lot. We will modify the corresponding expression when the formula is exactly the same as Dreamer. However, in our paper, to specifically demonstrate how the derived memory is generated, we denote the latent state $z={h_t,s_t}$ (which is similar to "PlaNet of the Bayesians and "Dreaming"), and Dreamer uses $s_{t}$ to represent the whole latent state. That is the reason why we say "similar to Dreamer," not" as in Dreamer."
>
>
> Q6: The relation of the analysis in 4.3 to [31] is not mentioned.
>
> Response:
>
> Based on [31], we give the value estimation error after adding the virtual trajectory generated by derived memory and reweighting technology. [31] gives a basic analysis framework of traditional model-based methods. Considering the limitation of pages, we mentioned the detailed relationships between our analysis and [31] in the supplemental materials. We will add a citation of [31] in our paper accordingly.
>
> Q7: How was the noise variance for the latent perturbation chosen? Was the regular transition noise variance similarly tuned?
>
> Response:
>
> The noise variance is a fixed hyper-parameter used in all environments. Specifically, the noise variance for the latent perturbation is 0.5. We took 10 groups of parameters in the (0,1) interval and finally found that 0.5 has a consistent improvement effect on most tasks. The regular transition noise variance is used as same as Dreamer and is determined by the learned state transition model.
>
>
> Q8: The notation around some concepts doesn’t help
>
> Clarification of notation and some concepts:
>
> The latent state at time $t$ is $z_{t}$ which consists of $z_{t} = (s_{t}, h_{t})$, where $s_{t}$ and $h_{t}$ are the probabilistic and deterministic variables respectively.
> The original/real memory is every encountered latent state in the replay buffer. And the derived memories are generated by performing a transformation on the original memory. These two expressions can not be interchanged with "real sequences" and "overshoot/fake sequences" in our paper.
>
> The real sequences
>
> $Seq_{\text {real}}=[z_{\tau}, a_{\tau}, z_{\tau+1}]_{\tau=t}^{t+T}$
>
> are the real trajectories the agent encountered and contain a series of latent states at different times.
>
> The overshoot/fake sequences
>
> $
> Seq_{\text{fake}}=[z_{t}, a_{t}, {\hat{z}}_{t+1}] \bigcup [\hat\{z}_\tau, a_\tau, \hat\{z}_\{\tau+1}]_\{\tau=t+1}^\{t+T}
> $
>
> are the trajectories predicted by the learned transition model.
> The input of the evaluator is $[z_{\tau}^{i}, a_{\tau}^{i}, z_{\tau+1}^{i}]$, where $i$ is the index of data, and the training of the evaluator aims to maximize
>
> $E_{(z_{\tau}, a_{\tau}) \sim D}$ {$\sum_{\tau=t}^{t+N} [ \log \left(\mathcal{E}\left(z_{\tau}, a_{\tau}, z_{\tau+1}\right)\right)+\log \left(1-\mathcal{E}\left(z_{\tau}, a_{\tau}, \hat{z}_{\tau+1}\right)\right)]$}.
>
>
>
> Finally, we hope we resolve all of your concerns and will be continued to polish our language and the clarity in our revision. Thanks again for your comments, and we sincerely wish you could reconsider your score.

---

> > ### Comment · Reviewer_S88K · 2021-08-26
> > **Further clarification.**
> >
> > Thank you for your detailed response and the new figure which helps.
> >
> > I think I'm starting to understand the notation around your evaluator and sequence of latents now, but I want to reiterate why I found it confusing in the original version:
> > - L140, you wrote $\hat\tau$ instead of $\hat{z}$        [now I understand it was a typo]
> > - Eq 3, the input to the evaluator is a sequence (also suggested by Fig2). In fact the evaluator $\mathcal{E}$ takes a latent transition (z,a,z') as input (either derived from the real or fake traj), as your new notation suggests. Is this understanding correct?

---

> > > ### Author Response · Authors · 2021-08-27
> > > **Thanks for your reply**
> > >
> > > To Reviewer S88K:
> > >
> > > Thanks for your reply and understanding. Sorry for the confusing statement in our previous version, your current understanding of the notation is correct. IDM divides a whole latent sequence into several $(z, a, z^{\prime})$ tuples as the input of the evaluator. We have revised Eq3 in the previous response(Q8). The reason that we did not draw such a specific process in detail is that Fig2 is a schematic diagram of the overall idea, we would like it to be intuitive and concise. In order to better elaborate such a process, we will add another figure with corresponding text in our final version appendix accordingly,  and you can see the figure on " https://sites.google.com/view/idm-evaluator ".  We sincerely thank you for your constructive suggestions.
> > >
> > > We hope we have resolved all the issues and showed the improved quality of the paper. And we deeply appreciate that if you could reconsider the score accordingly.
> > > We are always willing to address any of your further concerns.
> > >
> > > Thanks for your hard work. The authors.

---

> > > > ### Comment · Reviewer_S88K · 2021-08-28
> > > > **updated score**
> > > >
> > > > Thank you for confirming this. I've updated my score to reflect the improved clarity of the paper after the revisions.

---

> > > > > ### Author Response · Authors · 2021-08-29
> > > > > **Thank you for your responsible reply**
> > > > >
> > > > > We sincerely thank you for your responsible reply and understanding. We will continue to polish our paper by your suggestion. Thanks a lot!
> > > > >
> > > > > Thanks for your hard work. The authors.

---

> ### Author Response · Authors · 2021-08-19
> **Willing to answer any of your further concerns**
>
> We would like to thank you again for your responsible review and valuable suggestions. Your main criticism is largely about the imprecise/inconsistency that exists in our language. We do agree that the raised issues are important yet addressable, we would expect that these concerns have been largely fixed in this rebuttal session and we also have updated Fig 1 with the same top trajectories as your suggested (https://sites.google.com/view/idm-mbrl). Thus, we think these concerns may not outweigh the overall technical contributions of our work, and we have clarified most of them to make the paper clearer and rigorous.
>
>  **We are always willing to answer any of your further concerns** and we sincerely wish you to value the technical innovation and overall contributions of the paper as you also shed positive comments on such a perspective.

---

### Official Review · Reviewer_8jCq · 2021-07-16

**Rating:** 6
**Confidence:** 3

**Summary:**

This paper describes Imagining with Derived Memory (IDM), a model-based reinforcement learning method based on a world model. The proposed method is inspired by the memory prosthesis proposed by neuroscientists. The method introduces an additional disturbance into the latent variable of the recurrent state-space model (RSSM) to facilitate the robustness of the model. The experiment shows the IDM outperformed a baseline method, Dreamer, in several tasks.

**Limitations And Societal Impact:**

To improve the paper further, please consider the following comments.

(1) Contrastive learning
In world-model learning, contrastive learning-based approaches are gathering attention. They perform SOTA in many tasks, i.r., Dreamer has not already been SOTA. The paper refers to the topic just in conclusion and implicitly in 6.3. It's better to mention the approach in 6.3 explicitly.
Also, CVRL [1] is worth mentioning.

[1] Xiao Ma and Siwei Chen and David Hsu and Wee Sun Lee, Contrastive Variational Model-Based Reinforcement Learning for Complex Observations, Proceedings of the 4th Conference on Robot Learning (CoRL), 2020

The reference to Dreaming is missing.

[2] Masashi Okada, Tadahiro Taniguchi, Dreaming: Model-based Reinforcement Learning by Latent Imagination without Reconstruction, IEEE International Conference on Robotics and Automation (ICRA) 2021

(2) Relevancy between the latent states and the real physical states

In Section 3, the authors describe
"We add a decoding constraint to the RSSM, which improves relevancy
 between the derived latent states and the real physical states, thus avoiding the distortion after decoding."

However, the reason why the approach can improve the relevancy between the latent states and the "physical" states is not explained appropriately. The metric on images is different from the metric on a physical system in robot control.  I think there are no clear reasons and evidence that the proposed method improves the relevancy.

(3) Evaluation and baseline methods
The method is compared with Dreamer on six tasks alone.  Further evaluation employing other baseline methods, which the authors mention in Section 6, for example, is expected to show the effectiveness of the method.
Also, reference [2] is pointing out that Dreamer has an "object vanishing problem."  In the experiments in this paper, the tasks involving the "object vanishing problem" are omitted. The reason why they picked up the six tasks is described in 5.1. However, it will be fair to show the results of other tasks. To my understanding, the selection of tasks seems to be a bit biased.

(4)  Related work
The proposed method extended Dreamer by introducing diversity, i.e., sampling-based approximate distribution, at a latent pace to improve the robustness of the world model. From this viewpoint, the following paper is related.  I suggest the authors mention the study.

[3] Masashi Okada, Norio Kosaka, Tadahiro Taniguchi, PlaNet of the Bayesians: Reconsidering and Improving Deep Planning Network by Incorporating Bayesian Inference, IEEE/RSJ International Conference on Intelligent Robots and Systems (IROS) 2020


**Main Review:**

This paper describes Imagining with Derived Memory (IDM), a model-based reinforcement learning method based on a world model.
Generally, the paper is written properly and intuitively. The approach is understandable.
The theoretical analyses also add value to the paper.
As a whole, the paper has good originality, quality, and clarity.
However, the significance is not so high for two reasons.
1) The theoretical explanation about why and how the proposed method, IDM, contributes to the improvement of the performance is missing. Therefore, it is hard to recognize the theoretical contribution.
2)  The experiments are limited in tasks and analytical discussion is missing (i.e., they only show RL performance). The experiments is not sufficient enough to support all the authors' claims.
Also, the following points written in limitations should be considered.






**Time Spent Reviewing:**

6 hours

---

> ### Author Response · Authors · 2021-08-08
> **Response to Reviewer 8jCq**
>
> We thank you for your comments and advice, the detailed responses regrading each problem are listed below.
>
> Q1: The theoretical explanation about why and how the proposed method, IDM, contributes to the improvement of the performance is missing.
>
> Response:
>
> In Section 4.3, we introduce the theoretical analysis of the effectiveness of the algorithm and theoretically analyze the IDM modules' impact on the value estimation error.
> We give the upper bound of the value estimation error for MBRL under the IDM framework.
> $\epsilon_{w}$ indicates how well is the IDM helps to approximate the imagined distribution $p^{\prime}(z^{\prime}, r \mid z, a)$ to the real distribution $p(z^{\prime}, r \mid z, a)$.
>  The trajectories imagined with the derived memory increase the sample diversity, while the evaluator plays the role of importance sampling by reweighting the trajectories by $w(z)$. Coherent work of these two modules should effectively decrease the  $\epsilon_{w}$, thus reduces the estimation error of the value function, so as to learn a more accurate value net and further improve the performance of RL.
>
> Q2: Contrastive learning methods perform SOTA in many tasks, i.r., Dreamer has not already been SOTA.
>
> Response:
>
> We consider Dreamer as the SOTA of the model-based methods, not the SOTA of all kinds of RL methods. Thus, We also take DrQ (the SOTA of model-free methods) as a baseline to better support our conclusions.
> As your kindly mentioned, contrastive learning-based approaches are gathering increasing attention, it has been applied to learn latent world models [1,2,3,4] motivated from different perspectives. Hafner et al.[3] try to use contrastive learning as an alternative to image reconstruction, however, the contrastive learned agent gives worse performance compared with the one learned by image reconstruction. Dreaming[4] derives an InfoMax objective of contrastive learning from the evidence lower bound of Dreamer and removes the decoder to solve the object vanishing problem of the representation methods based on image reconstruction. CVRL[1] emphasizes the strength of contrastive learning in handling complex visual observations. However, CVRL is not a pure model-based method and uses a hybrid training scheme that includes a model-based scheme (Dreamer) and a model-free scheme (SAC), as mentioned in CVRL’s Section 3.3. And CVRL only claims that it achieves comparable performance on the DMControl with the SOTA method Dreamer (on page 6), i.e., its performance does not exceed Dreamer in DMControl tasks, and CVRL also considers Dreamer as the SOTA of model-based methods in DMControl tasks.
> Furthermore, the key idea of IDM is not in conflict with contrastive learning. They aim to improve performance from different perspectives. Since our method is compatible with representation learning techniques, we can extend our framework by incorporating advanced contrastive representation learning to further improve the overall performance, which is left as future work.
>
>
> [1] Xiao Ma and Siwei Chen and David Hsu and Wee Sun Lee, Contrastive Variational Model-Based Reinforcement Learning for Complex Observations, Proceedings of the 4th Conference on Robot Learning (CoRL), 2020.
>
> [2] T. Kipf, E. van der Pol, and M. Welling. Contrastive learning of structured world models. In
> International Conference on Learning Representations, 2020.
>
> [3] D. Hafner, T. Lillicrap, J. Ba, and M. Norouzi. Dream to control: Learning behaviors by latent
> imagination. arXiv preprint arXiv:1912.01603, 2019.
>
> [4] Masashi Okada, Tadahiro Taniguchi, Dreaming: Model-based Reinforcement Learning by Latent Imagination without Reconstruction, IEEE International Conference on Robotics and Automation (ICRA) 2021
>
> Q3: The reference to Dreaming is missing
>
> Response:
>
>  Thanks for your suggestion. We added reference to Dreaming in the discussion of future work.
>
> Q4: the reason why the approach can improve the relevancy between the latent states and the "physical" states is not explained.The metric on images is different from the metric on a physical system in robot control.
>
> Response:
>
> There may be some ambiguities in the expression of the relevancy between the latent states and the physical states. Here, it refers to the probability that the decoded image could be photographed from a real existing state of the robot rather than the relationship between latent state and physical quantities with specific meaning (such as the height of the center of gravity of the object, etc.) Image is the observation of physical state. If the observation is distorted, it indicates that the information of physical state is greatly missing.
>
> Q5: The experiments are limited in tasks, the tasks involving the "object vanishing problem" are omitted.
>
> Response:
>
> Different from the major contribution of dreaming, this paper focuses on improving sample efficiency and the policy robustness to image uncertainty.
> Both Dreamer and IDM learn latent representation based on reconstruction, and as claims in Dreaming, "object vanishing" is a common problem of the methods relying on reconstruction, thus they would not perform much differently in the "object vanishing critical" tasks given in Dreaming paper.
> Besides, these tasks are also rarely used for providing core experimental results in other critical papers such as CURL, DrQ, and as also Dreamer.
>
> Q6: The analytical discussion is missing (i.e., they only show RL performance).
>
> Response:
>
>  This paper mainly aims to improve the model usage of the model-based method. We build our world model based on the RSSM, which is also used in Dreamer and PlaNet. We achieve comparable model prediction performance with Dreamer, and more analytical discussions on video prediction by the world model we used are shown in Appendix A.3. We also make a detailed discussion about policy robustness and sample efficiency in Section 5.1.
>
>
> Q7: The proposed method extended Dreamer by introducing diversity, i.e., sampling-based approximate distribution, at a latent space to improve the robustness of the world model. I suggest the authors mention the study "PlaNet of the Bayesians"
>
> Response:
>
> Thanks for your suggestion. We will add it to the related works.
> "PlaNet of the Bayesians" represents the model uncertainty by a neural network ensemble to approximately infer model posteriors and further improves the performance of PlaNet. However, IDM and "PlaNet of the Bayesians" improves RL performance from different perspectives. IDM aims to improve the policy robustness to the uncertainty of input images by enable the agent to encounter diverse latent states, while "PlaNet of the Bayesians" aims to reduce the influence of the model prediction error by model ensemble method. Moreover, there is no official open-source code given by the authors of "PlaNet of the Bayesians."

---

> > ### Comment · Reviewer_8jCq · 2021-08-25
> > **Response to the authors**
> >
> > Thank you very much for your response.
> > The responses are clear to me.
> > I hope the authors update the paper considering my and other reviewers' comments.

---

> > ### Author Response · Authors · 2021-08-25
> > **Thank you for your reply and understanding**
> >
> > Thank you for your reply and understanding.  We will update our paper considering your and other reviewers' comments accordingly. We sincerely thank you for your comprehensive comments.

---

### Official Review · Reviewer_EMUa · 2021-07-20

**Rating:** 6
**Confidence:** 3

**Summary:**

This paper presents a model-based reinforcement learning agent that uses imagined trajectories, obtained via rollouts from noised initial states observed by the agent, to improve data efficiency and robustness. The approach builds on top of Dreamer, a popular MBRL approach that trains a latent dynamics model and trains a policy via value gradients computed on imagined rollouts from the latent model in an actor-critic setup. The presented approach, called Imagining with Derived Memory (IDM), improves upon Dreamer by generating diverse imagined rollouts; this is done by adding random perturbations to the latent state of the initial observation from which an imagined rollout can be derived. By augmenting the training data of standard imagined rollouts (whose initial states were observed in the environment) with derived rollouts from these perturbed initial states the approach generates diverse additional data to effectively double the batch size.

To ensure that the derived rollouts are physically plausible two additional changes are made to Dreamer. First, a decoding constraint is added to the model’s training loss; this loss constrains the reconstructions from the original and perturbed latents to remain close to each other, thereby encouraging the perturbed latents to also encode plausible latent states. Second, to ensure that only physically plausible rollouts are used for training, the approach additionally trains a discriminator that scores each trajectory; this discriminator is trained to distinguish between real and imagined trajectories and in practice scores “real-looking”, and therefore more plausible trajectories higher than implausible ones. This score is used to weight the loss for training the policy and value function, thereby mitigating the effect of implausible trajectories affecting learning. Additionally, a theoretical analysis of the IDM framework is also provided which upper bounds the error between true returns and returns achieved in the model.

The approach is tested on six tasks in the control suite, both with and without observation uncertainty (random image rotations within +- 5 degrees) and compared with Dreamer and DrQ, a state of the art model-free approach. IDM performs better than both baselines on 3/6 tasks without observation uncertainty, and significantly outperforms both baselines on all tasks with observation uncertainty added. A further study that ablates different parts of the algorithm including the use of derived rollouts, the use of the discriminator for scoring rollouts and the decoding constraint is provided; in general the combination of all three is needed for improving data efficiency in comparison to baselines.

**Limitations And Societal Impact:**

The paper does discuss some limitations of the proposed approach but a better discussion of the strengths and weaknesses are needed and can add value to the paper; adding some directions for future work would also be helpful. Societal impact is not addressed.

**Main Review:**

Originality: The approach builds upon prior work, primarily Dreamer, with the key addition being the use of imagined rollouts from noisy latent states to augment the training data. Other additions such as the decoding constraint and discriminator for scoring trajectories are needed to stabilise learning when using these derived rollouts. The paper does a good job of establishing related work and connections to prior work. I find the general idea of augmenting training data with diverse imagined rollouts quite interesting and useful,  but there are several open questions that could have been addressed.

1. The approach only tries a simple way of doing this which is to add random Gaussian perturbations to the initial state latents. Are there other ways in which diverse imagined trajectories could be generated? Are there different perturbations that could be applied (e.g. correlated perturbations in different directions in the latent space)? It would have been nice to see some comparisons or a discussion on this topic.
2. Similarly, could the decoding constraint be made proportional to the amount of noise added? i.e. latents that are close in magnitude need to have a stronger KL constraint than those that are perturbed more heavily.

Quality & Clarity: IDM shows significant improvements in performance upon Dreamer and DrQ on the control suite tasks with observation uncertainty and similar or slightly lower performance on some tasks without uncertainty. The ablation study also provides some interesting insights; the fact that the scoring of trajectories is primarily needed for complex tasks such as the hopper where the model can struggle to produce plausible rollouts compared to walker-walk where the dynamics are more straightforward is a very useful insight. The theoretical analysis mostly mirrors that of Janner et al. with the addition of the weight from the discriminator providing a slightly different bound. While the results are reasonable the paper could be significantly strengthened further by running a few additional experiments and adding some discussion to provide further insight. The writing could also be further improved to improve consistency and clarity. A few questions/suggestions:
1. IDM seems to use a very long imagination horizon (H=50) for the control tasks, which is also far larger than the sequence length (L=15) on which the model was trained. Dreamer uses H=15 (from the paper). Is there a reason why this horizon was needed? Would it not hurt to use a horizon much longer than the sequence length as it might result in spurious gradients (even with the lambda returns). It would also be useful to see an experiment where the horizon is varied; does the horizon matter for achieving good performance?
2. What is the architecture of the MLP used as the discriminator? Does it concat all observations from all time steps to compute a score? Is there a different discriminator for each trajectory length (since Dreamer uses sub-trajectory returns)?
3. Is there a specific reason why the random rotation perturbation was chosen? What about other forms of perturbations? Results on an additional form of perturbation would add more strength to the paper.
4. What is the form of observation uncertainty used for DrQ in Figure 4? Is this the same random rotation for each frame in the stack or different random rotations?
5. A more pertinent comparison w.r.t observation uncertainty would be RAD; RAD is trained with different image augmentations (including rotations). It would be interesting to see how well that performs to get a baseline number.
6. A key advantage of IDM is that it can be used to generate significant additional diverse imaginations. The paper only explores generating a single additional imagined rollout per state, i.e. the batch size is increased by a factor of 2. Is there any reason not to generate significantly more data? It would be great to see an experiment varying the effective batch size (e.g. 1 (no additional imagined data), 2 (current IDM), 4 (3x derived trajectories, 1x standard imagined trajs), 8 (7x/1x derived/standard imagined trajs) and so on. One would expect to see some gains for further increase in imagined samples up to a point.
7. In terms of clarity, the writing could be further improved (e.g. Eqn 2 can be better phrased to show that a_t is first sampled and used to generate h_t+1, s_t+1, and a for loop can be added to show the full imagined rollout). The notation can be made more consistent (e.g. p_\theta is used in the main text interchangeably with p). Both these would improve readability.

Significance: The approach presented in this paper provides an interesting way to augment training data with diverse imagined rollouts to improve learning efficiency and robustness to perturbations. While the initial results are promising, more experiments, ablations and discussions are needed to further strengthen the paper before the results can be attractive to the broader research community.

**Time Spent Reviewing:**

6

---

> ### Author Response · Authors · 2021-08-08
> **Response to Reviewer EMUa**
>
> We sincerely thank you for your comprehensive comments on our paper and we carefully answer each of your questions as below.
>
> Q1: The approach only tries a simple way of doing this, which is to add random Gaussian perturbations to the initial latent state. Are there other ways in which diverse imagined trajectories could be generated?
> Are there different perturbations that could be applied (e.g., correlated perturbations in different directions in the latent space)?
>
> Response:
>
> The Gaussian noise in latent space could be theoretically mapped to any types of uncertainty in image level, and this principle is extensively used in VAE and GAN. Adding Gaussian perturbations should be an intuitive way to simulate the influences in latent space caused by the uncertainty of input image, since
> the stochastic part of latent state $s_{t}$ in RSSM obeys Gaussian distribution, any image will be encoded to a Gaussian variable in latent space under the RSSM framework. Besides, it naturally reduces the difficulty of solving the decoding constraint.
> Advanced perturbations should further enhance the diversity of derived memory, which further expands the latent sample space in policy learning.
>
> And thanks for your constructive advice, adding correlated perturbations could be very promising to be studied. One thing that needs to be noticed is that the covariance metric of correlated noise is high dimensional and complex to assign since the latent state contains $h_{t}$ with 200 dimensions and $s_{t}$ with 30 dimensions. It will also increase the complexity for solving the decoding constraint, which is a problem worth investigating. Furthermore, approaches for derived memory generation with considering task/episodic transfer would also be researched as our future works.
>
> Q2: Could the decoding constraint be made proportional to the amount of noise added?
>
> Response:
>
> Yes, the derived memory depends on the decoding constraint. If we want to add noise with a larger variance, the constraint should be stronger. However, since we use the penalty method to solve the contained problem, a too strong constraint will damage the learning process of the transition and representation model.
>
> Q3: IDM seems to use a very long imagination horizon (H=50) for the control tasks
>
> Response:
>
> Sorry for the misleading statement, it is a typo. The imagination horizon (H=15) and sequence length (L=50) are the same in Dreamer. We will revise Table 3 in Appendix A.4 accordingly.
>
>
> Q4: What is the architecture of the MLP used as the discriminator? Does it concat all observations from all the time steps to compute a score? Is there a different discriminator for each trajectory length (since Dreamer uses sub-trajectory returns)?
>
> Response:
>
> The input of the MLP is the concatenation of $[z_t,a_t,z_{t+1}]$, the output of the MLP will be sent to the softmax, and then the final score is obtained.  Considering the fact that the Dreamer uses sub-trajectory returns (the length may be different) and the derived memory is used as the start point of the imagined trajectory, we use the transition predict the score of $z_t,a_t,z_{t+1}$ as the weight of sub-trajectory.
>
>
> Q5: Is there a specific reason why the random rotation perturbation was chosen? What about other forms of perturbations?
>
> Response:
>
> We choose the random rotation perturbation to simulate the random swing caused by the robot's motion when the camera's fixed device is loose. Random rotation is a representative uncertainty the agent may encounter in the real world. During the training process of RL, images are only collected horizontally from the simulator, but in the real world, the camera could be skewed in a different direction, which changes the angles that agents or objects appear in the input image. Results on additional forms of perturbation (such as random crop and horizontal flip) will be added and We are running related supplementary experiments now.
>
>
>
> Q6: What is the form of observation uncertainty used for DrQ in Figure 4?
>
> Response:
>
> It is the same random rotation for each frame in the stack.
>
>
>
> Q7: Is there any reason not to generate significantly more data?
>
> Response:
>
> Actually, generating more data will indeed promote policy learning.
> However, generating data also will increase the occupation of GPU memory.
> Considering the computational efficiency and the constraints of the computational resources, conducting single sampling when generating derived memory is appropriate and effective.
>
> Thanks again for reading our article carefully and giving very constructive suggestions. We hope we resolve all of your concerns and we wish you could reconsider your score.

---

### Decision · Program_Chairs · 2021-09-27

**Decision:**

Accept (Poster)

**Comment:**

The submission introduces Imagining with Derived Memory (IDM), a novel extension of the Dreamer agent, a model based RL methodology using rollouts in latent space to improve policy training.
The proposed method wants to improve both training efficiency and robustness, and at its core involves regularising the dynamic model by explicitly constraining the imagined trajectories to be smoother wrt to perturbation of the latents, i.e. perturbed latents should map to similar reconstructions.  The smooth model is used to generate a richer set of trajectories, compared to Dreamer, by perturbing initial states collected interacting with the environment, and reweighting them in the final training loss using a learned discriminator function, ensuring plausibility of the sampled trajectories.

I expect the new elements presented in this paper to be of broad interest to the community; in particular, the way training data is augmented is widely applicable, and could motivate more groups to refine the presented technique, addressing different sources of uncertainty in the trajectory data.
One drawback of the paper it that the experimental section only presents limited (although quite promising!) results on continuous control tasks; while more experiments and ablations would make the paper more immediately impactful, reviewers agreed that the current experimental setup is sufficient to support the claims made in the submission.

Finally, the reviewers made numerous suggestions on how to improve the presentation of both the methods and results, and authors have already incorporated feedback, or committed to include it in the next iteration of the paper.